# Compositional Explanations of Neurons

**Jesse Mu**
Stanford University
muj@stanford.edu

**Jacob Andreas**
MIT CSAIL
jda@mit.edu

## Abstract

We describe a procedure for explaining neurons in deep representations by identifying *compositional* logical concepts that closely approximate neuron behavior. Compared to prior work that uses atomic labels as explanations, analyzing neurons compositionally allows us to more precisely and expressively characterize their behavior. We use this procedure to answer several questions on interpretability in models for vision and natural language processing. First, we examine the kinds of abstractions learned by neurons. In image classification, we find that many neurons learn highly abstract but semantically coherent visual concepts, while other *polysemantic* neurons detect multiple unrelated features; in natural language inference (NLI), neurons learn shallow lexical heuristics from dataset biases. Second, we see whether compositional explanations give us insight into model performance: vision neurons that detect human-interpretable concepts are positively correlated with task performance, while NLI neurons that fire for shallow heuristics are negatively correlated with task performance. Finally, we show how compositional explanations provide an accessible way for end users to produce simple "copy-paste" adversarial examples that change model behavior in predictable ways.

## 1   Introduction

In this paper, we describe a procedure for automatically explaining logical and perceptual abstractions encoded by individual neurons in deep networks. Prior work in neural network interpretability has found that neurons in models trained for a variety of tasks learn human-interpretable concepts, e.g. faces or parts-of-speech, often without explicit supervision [5, 10, 11, 27]. Yet many existing interpretability methods are limited to ad-hoc explanations based on manual inspection of model visualizations or inputs [10, 26, 27, 35, 38, 39]. To instead automate explanation generation, recent work [5, 11] has proposed to use labeled "probing datasets" to explain neurons by identifying concepts (e.g. *dog* or *verb*) closely aligned with neuron behavior.

However, the atomic concepts available in probing datasets may be overly simplistic explanations of neurons. A neuron might robustly respond to images of dogs without being exclusively specialized for dog detection; indeed, some have noted the presence of *polysemantic* neurons in vision models that detect multiple concepts [12, 27]. The extent to which these neurons have learned meaningful perceptual abstractions (versus detecting unrelated concepts) remains an open question. More generally, neurons may be more accurately characterized not just as simple detectors, but rather as operationalizing complex decision rules composed of multiple concepts (e.g. *dog faces, cat bodies, and car windows*). Existing tools are unable to surface such compositional concepts automatically.

We propose to generate explanations by searching for logical forms defined by a set of composition operators over primitive concepts (Figure 1). Compared to previous work [5], these explanations serve as better approximations of neuron behavior, and identify behaviors that help us answer a variety of interpretability questions across vision and natural language processing (NLP) models. First, what kind of logical concepts are learned by deep models in vision and NLP? Second, do the

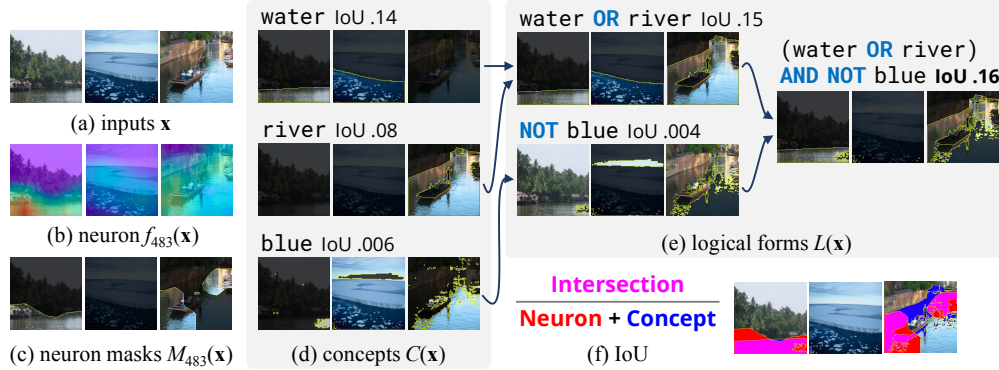

Figure 1: Given a set of inputs (a) and scalar neuron activations (b) converted into binary masks (c), we generate an explanation via beam search, starting with an inventory of primitive concepts (d), then incrementally building up more complex logical forms (e). We attempt to maximize the IoU score of an explanation (f); depicted is the IoU of $M_{483}(\mathbf{x})$ and (`water OR river`) `AND NOT blue`.

quality and interpretability of these learned concepts relate to model performance? Third, can we use the logical concepts encoded by neurons to control model behavior in predictable ways? We find that:

1. Neurons learn compositional concepts: in **image classification**, we identify neurons that learn meaningful perceptual abstractions (e.g. *tall structures*) and others that fire for unrelated concepts. In natural language inference (**NLI**), we show that shallow heuristics (based on e.g. gender and lexical overlap) are not only learned, but reified in individual neurons.

2. Compositional explanations help predict model accuracy, but interpretability is not always associated with accurate classification: in **image classification**, human-interpretable abstractions are *correlated* with model performance, but in **NLI**, neurons that reflect shallower heuristics are *anticorrelated* with performance.

3. Compositional explanations allow users to predictably manipulate model behavior: we can generate crude "copy-paste" adversarial examples based on inserting words and image patches to target individual neurons, in contrast to black-box approaches [1, 36, 37].

## 2   Generating compositional explanations

Consider a neural network model $f$ that maps inputs $\mathbf{x}$ to vector representations $r \in \mathbb{R}^d$. $f$ might be a prefix of a convolutional network trained for image classification or a sentence embedding model trained for a language processing task. Now consider an individual neuron $f_n(\mathbf{x}) \in \mathbb{R}$ and its activation on a set of concrete inputs (e.g. ResNet-18 [15] layer 4 unit 483; Figure 1a–b). How might we explain this neuron's behavior in human-understandable terms?

The intuition underlying our approach is shared with the NetDissect procedure of Bau et al. [5]; here we describe a generalized version. The core of this intuition is that a good explanation is a *description* (e.g. a named category or property) that identifies the same inputs for which $f_n$ activates. Formally, assume we have a space of pre-defined atomic *concepts* $C \in \mathcal{C}$ where each concept is a function $C : \mathbf{x} \mapsto \{0, 1\}$ indicating whether $\mathbf{x}$ is an instance of $C$. For image pixels, concepts are image segmentation masks; for the *water* concept, $C(\mathbf{x})$ is 1 when $\mathbf{x}$ is an image region containing water (Figure 1d). Given some measure $\delta$ of the similarity between neuron activations and concepts, NetDissect explains the neuron $f_n$ by searching for the concept $C$ that is most similar:

$$\textsc{Explain-NetDissect}(n) = \underset{C \in \mathcal{C}}{\arg\max}\, \delta(n, C). \tag{1}$$

While $\delta$ can be arbitrary, Bau et al. [5] first *threshold* the continuous neuron activations $f_n(\mathbf{x})$ into binary masks $M_n(\mathbf{x}) \in \{0, 1\}$ (Figure 1c). This can be done *a priori* (e.g. for post-ReLU activations, thresholding above 0), or by dynamically thresholding above a neuron-specific percentile. We can then compare binary neuron masks and concepts with the Intersection over Union score (IoU, or Jaccard similarity; Figure 1f):

$$\delta(n, C) \triangleq \mathrm{IoU}(n, C) = \Big[ \sum_{\mathbf{x}} \mathbb{1}(M_n(\mathbf{x}) \wedge C(\mathbf{x})) \Big] \Big/ \Big[ \sum_{\mathbf{x}} \mathbb{1}(M_n(\mathbf{x}) \vee C(\mathbf{x})) \Big]. \tag{2}$$

**Compositional search.** The procedure described in Equation 1 can only produce explanations from the fixed, pre-defined concept inventory $\mathcal{C}$. Our main contribution is to combinatorially expand the set of possible explanations to include *logical forms* $\mathcal{L}(\mathcal{C})$ defined inductively over $\mathcal{C}$ via composition operations such as disjunction (OR), conjunction (AND), and negation (NOT), e.g. $(L_1 \text{ AND } L_2)(\mathbf{x}) = L_1(\mathbf{x}) \wedge L_2(\mathbf{x})$ (Figure 1e). Formally, if $\Omega_\eta$ is the set of $\eta$-ary composition functions, define $\mathcal{L}(\mathcal{C})$:

1. Every primitive concept is a logical form: $\forall C \in \mathcal{C}$, we have $C \in \mathcal{L}(\mathcal{C})$.
2. Any composition of logical forms is a logical form: $\forall \eta, \ \omega \in \Omega_\eta, \ (L_1, \ldots, L_\eta) \in \mathcal{L}(\mathcal{C})^\eta$, where $\mathcal{L}(\mathcal{C})^\eta$ is the set of $\eta$-tuples of logical forms in $\mathcal{L}(\mathcal{C})$, we have $\omega(L_1, \ldots, L_\eta) \in \mathcal{L}(\mathcal{C})$.

Now we search for the best logical form $L \in \mathcal{L}(\mathcal{C})$:

$$\text{EXPLAIN-COMP}(n) = \underset{L \in \mathcal{L}(\mathcal{C})}{\arg\max} \, \text{IoU}(n, L). \tag{3}$$

The $\arg\max$ in Equation 3 ranges over a structured space of compositional expressions, and has the form of an inductive program synthesis problem [23]. Since we cannot exhaustively search $\mathcal{L}(\mathcal{C})$, in practice we limit ourselves to formulas of maximum length $N$, by iteratively constructing formulas from primitives via beam search with beam size $B = 10$. At each step of beam search, we take the formulas already present in our beam, compose them with new primitives, measure IoU of these new formulas, and keep the top $B$ new formulas by IoU, as shown in Figure 1e.

## 3 Tasks

The procedure we have described above is model- and task-agnostic. We apply it to two tasks in vision and NLP: first, we investigate a scene recognition task explored by the original NetDissect work [5], which allows us to examine compositionality in a task where neuron behavior is known to be reasonably well-characterized by atomic labels. Second, we examine *natural language inference* (NLI): an example of a seemingly challenging NLP task which has recently come under scrutiny due to models' reliance on shallow heuristics and dataset biases [13, 14,

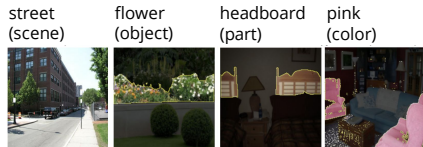

whether compositional explanations can uncover such undesirable be

**Image Classification.** NetDissect [5] examines whether a convolutional neural network trained on a scene recognition task has learned detectors that correspond to meaningful abstractions of objects. We take the final 512-unit convolutional layer of a ResNet-18 [15] trained on the Places365 dataset [40], probing for concepts in the ADE20k scenes dataset [41] with atomic concepts defined by annotations in the Broden dataset [5]. There are 1105 unique concepts in ADE20k, categorized by Scene, Object, Part, and Color (see Figure 2 for examples).

Figure 2: Example concepts from the Broden dataset (reproduced with permission).

Broden has pixel-level annotations, so for each input image $\mathbf{X} \in \mathbb{R}^{H \times W}$, inputs are indexed by pixels $(i, j)$: $\mathbf{x}_{i,j} \in \mathcal{X}$. Let $f_n(\mathbf{x}_{i,j})$ be the activation of the $n$th neuron at position $(i, j)$ of the image $\mathbf{X}$, after the neuron's activation map has been bilinearly upsampled from layer dimensions $H_l \times W_l$ to the segmentation mask dimensions $H \times W$. Following [5], we create neuron masks $M_n(x)$ via dynamic thresholding: let $T_n$ be the threshold such that $P(f_n(\mathbf{x}) > T_n) = 0.005$ over all inputs $\mathbf{x} \in \mathcal{X}$. Then $M_n(\mathbf{x}) = \mathbb{1}(f_n(\mathbf{x}) > T_n)$. For composition, we use operations AND ($\wedge$), OR ($\vee$), and NOT ($\neg$), leaving more complex operations (e.g. relations like ABOVE and BELOW) for future work.

**NLI.** Given premise and hypothesis sentences, the task of NLI is to determine whether the premise *entails* the hypothesis, *contradicts* it, or neither (*neutral*). We investigate a BiLSTM baseline architecture proposed by [7]. A bidirectional RNN encodes both the premise and hypothesis to form 512-d representations. Both representations, and their elementwise product and difference, are then concatenated to form a 2048-d representation that is fed through a multilayer perceptron (MLP) with two 1024-d layers with ReLU nonlinearities and a final softmax layer. This model is trained on the Stanford Natural Language Inference (SNLI) corpus [6] which consists of 570K sentence pairs.

Neuron-level explanations of NLP models have traditionally analyzed how RNN hidden states detect word-level features as the model passes over the input sequence [4, 10], but in most NLI models, these

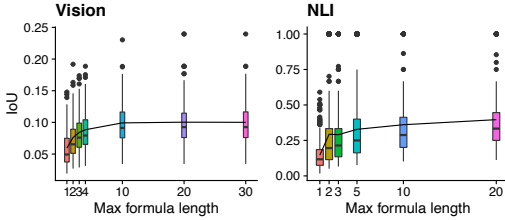

Figure 3: Distribution of IoU versus max formula length. The line indicates mean IoU. $N = 1$ is equivalent to NetDissect [5]; IoU scores steadily increase as max formula length increases.

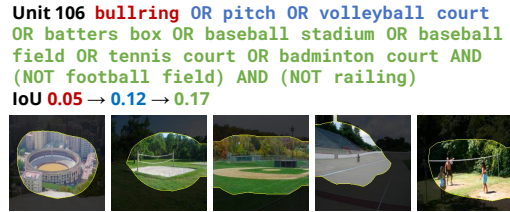

Figure 4: NetDissect [5] assigns unit 106 the label `bullring`, but in reality it is detects general sports fields, except football fields, as revealed by the **length 3** and **length 10** explanations.

RNN features are learned early and are often quite distant from the final sentence representation used for prediction. Instead, we analyze the MLP component, probing the 1024 neurons of the penultimate hidden layer for sentence-level explanations, so our inputs $\mathbf{x}$ are premise-hypothesis pairs.

We use the SNLI validation set as our probing dataset (10K examples). As our features, we take the Penn Treebank part of speech tags (labeled by SpaCy[1]) and the 2000 most common words appearing in the dataset. For each of these we create 2 concepts that indicate whether the word or part-of-speech appears in the premise or hypothesis. Additionally, to detect whether models are using lexical overlap heuristics [25], we define 4 concepts indicating that the premise and hypothesis have more than 0%, 25%, 50%, or 75% overlap, as measured by IoU between the unique words.

For our composition operators, we keep AND, OR, and NOT; in addition, to capture the idea that neurons might fire for groups of words with similar meanings, we introduce the unary NEIGHBORS operator. Given a word feature $C$, let the *neighborhood* $\mathcal{N}(C)$ be the set of 5 closest words $C'$ to $C$, as measured by their cosine distance in GloVe embedding space [28]. Then, $\text{NEIGHBORS}(C)(\mathbf{x}) = \bigvee_{C' \in \mathcal{N}(C)} C'(\mathbf{x})$ (i.e. the logical OR across all neighbors). Finally, since these are post-ReLU activations, instead of dynamically thresholding we simply define our neuron masks $M_n(\mathbf{x}) = \mathbb{1}(f_n(\mathbf{x}) > 0)$. There are many "dead" neurons in the model, and some neurons fire more often than others; we limit our analysis to neurons that activate reliably across the dataset, defined as being active at least 500 times (5%) across the 10K examples probed.

## 4 Do neurons learn compositional concepts?

**Image Classification.** Figure 3 (left) plots the distribution of IoU scores for the best concepts found for each neuron as we increase the maximum formula length $N$. When $N = 1$, we get EXPLAIN-NETDISSECT, with a mean IoU of 0.059; as $N$ increases, IoU increases up to 0.099 at $N = 10$, a statistically significant 68% increase ($p = 2 \times 10^{-9}$). We see diminishing returns after length 10, so we conduct the rest of our analysis with length 10 logical forms. The increased explanation quality suggests that our compositional explanations indeed detect behavior beyond simple atomic labels: Figure 4 shows an example of a *bullring* detector which is actually revealed to detect fields in general.

We can now answer our first question from the introduction: are neurons learning meaningful abstractions, or firing for unrelated concepts? Both happen: we manually inspected a random sample of 128 neurons in the network and their length 10 explanations, and found that **69%** learned some meaningful combination of concepts, while **31%** were *polysemantic*, firing for at least some unrelated concepts. The 88 "meaningful" neurons fell into 3 categories (examples in Figure 5; more in Appendix C; Appendix A.1 reports concept uniqueness and granularity across formula lengths):

1. 50 (57%) learn a perceptual **abstraction** that is also lexically coherent, in that the primitive words in the explanation are semantically related (e.g. to *towers* or *bathrooms*; Figure 5a).

2. 28 (32%) learn a perceptual **abstraction** that is *not* lexically coherent, as the primitives are not obviously semantically related. For example, `cradle OR autobus OR fire escape` is a vertical rails detector, but we have no annotations of vertical rails in Broden (Figure 5b).

3. 10 (12%) have the form $L_1$ AND NOT $L_2$, which we call **specialization**. They detect more specific variants of Broden concepts (e.g. (`water OR river`) AND NOT `blue`; Figure 5c).

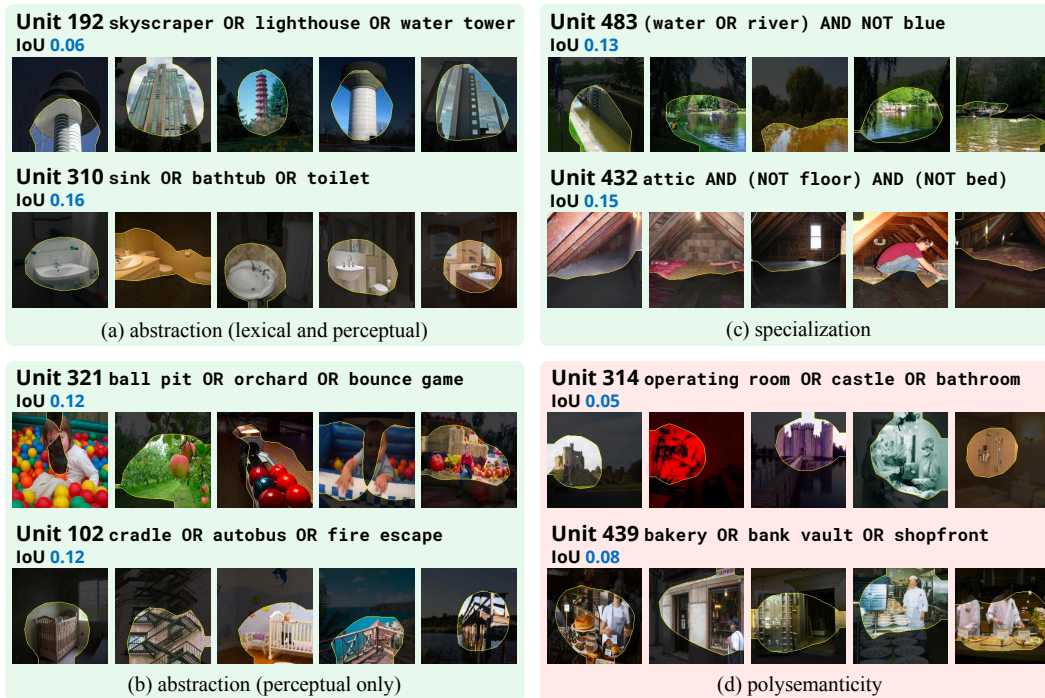

**Unit 192** `skyscraper OR lighthouse OR water tower`
**IoU 0.06**

**Unit 310** `sink OR bathtub OR toilet`
**IoU 0.16**

(a) abstraction (lexical and perceptual)

**Unit 483** `(water OR river) AND NOT blue`
**IoU 0.13**

**Unit 432** `attic AND (NOT floor) AND (NOT bed)`
**IoU 0.15**

(c) specialization

**Unit 321** `ball pit OR orchard OR bounce game`
**IoU 0.12**

**Unit 102** `cradle OR autobus OR fire escape`
**IoU 0.12**

(b) abstraction (perceptual only)

**Unit 314** `operating room OR castle OR bathroom`
**IoU 0.05**

**Unit 439** `bakery OR bank vault OR shopfront`
**IoU 0.08**

(d) polysemanticity

Figure 5: Image classification explanations categorized by **semantically coherent** abstraction (a–b) and specialization (c), and **unrelated** polysemanticity (d). For clarity, logical forms are length $N = 3$.

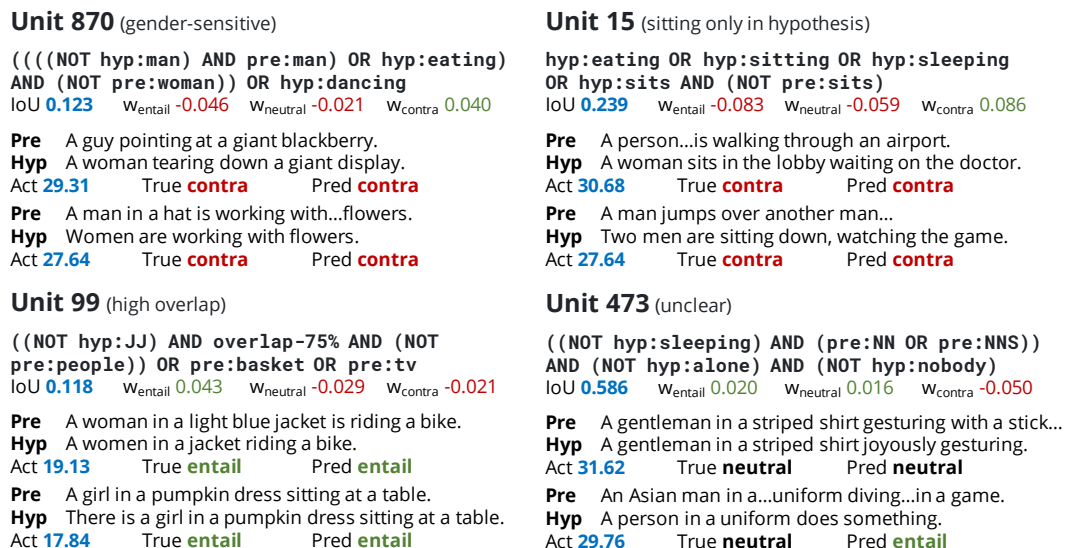

**Unit 870** (gender-sensitive)

`(((((NOT hyp:man) AND pre:man) OR hyp:eating) AND (NOT pre:woman)) OR hyp:dancing`
IoU **0.123**    $w_{entail}$ -0.046    $w_{neutral}$ -0.021    $w_{contra}$ 0.040

**Pre** A guy pointing at a giant blackberry.
**Hyp** A woman tearing down a giant display.
Act **29.31**    True **contra**    Pred **contra**
**Pre** A man in a hat is working with...flowers.
**Hyp** Women are working with flowers.
Act **27.64**    True **contra**    Pred **contra**

**Unit 99** (high overlap)

`((NOT hyp:JJ) AND overlap-75% AND (NOT pre:people)) OR pre:basket OR pre:tv`
IoU **0.118**    $w_{entail}$ 0.043    $w_{neutral}$ -0.029    $w_{contra}$ -0.021

**Pre** A woman in a light blue jacket is riding a bike.
**Hyp** A women in a jacket riding a bike.
Act **19.13**    True **entail**    Pred **entail**
**Pre** A girl in a pumpkin dress sitting at a table.
**Hyp** There is a girl in a pumpkin dress sitting at a table.
Act **17.84**    True **entail**    Pred **entail**

**Unit 15** (sitting only in hypothesis)

`hyp:eating OR hyp:sitting OR hyp:sleeping OR hyp:sits AND (NOT pre:sits)`
IoU **0.239**    $w_{entail}$ -0.083    $w_{neutral}$ -0.059    $w_{contra}$ 0.086

**Pre** A person...is walking through an airport.
**Hyp** A woman sits in the lobby waiting on the doctor.
Act **30.68**    True **contra**    Pred **contra**
**Pre** A man jumps over another man...
**Hyp** Two men are sitting down, watching the game.
Act **27.64**    True **contra**    Pred **contra**

**Unit 473** (unclear)

`((NOT hyp:sleeping) AND (pre:NN OR pre:NNS)) AND (NOT hyp:alone) AND (NOT hyp:nobody)`
IoU **0.586**    $w_{entail}$ 0.020    $w_{neutral}$ 0.016    $w_{contra}$ -0.050

**Pre** A gentleman in a striped shirt gesturing with a stick...
**Hyp** A gentleman in a striped shirt joyously gesturing.
Act **31.62**    True **neutral**    Pred **neutral**
**Pre** An Asian man in a...uniform diving...in a game.
**Hyp** A person in a uniform does something.
Act **29.76**    True **neutral**    Pred **entail**

Figure 6: NLI length 5 explanations. For each neuron, we show the explanation (e.g. `pre:x` indicates x appears in the premise), IoU, class weights $w_{\{entail,neutral,contra\}}$, and activations for 2 examples.

The observation that IoU scores do not increase substantially past length 10 corroborates the finding of [12], who also note that few neurons detect more than 10 unique concepts in a model. Our procedure, however, allows us to more precisely characterize whether these neurons detect abstractions or unrelated disjunctions of concepts, and identify more interesting cases of behavior (e.g. *specialization*). While composition of Broden annotations explains a majority of the abstractions learned, there is still considerable unexplained behavior. The remaining behavior could be due to noisy activations, neuron misclassifications, or detection of concepts absent from Broden.

**NLI.** NLI IoU scores reveal a similar trend (Figure 3, right): as we increase the maximum formula length, we account for more behavior, though scores continue increasing past length 30. However, short explanations are already useful: Figure 6, Figure 9 (explained later), and Appendix D show example length 5 explanations, and Appendix A.2 reports on the uniqueness of these concepts across formula lengths. Many neurons correspond to simple decision rules based mostly on lexical features: for example, several neurons are *gender sensitive* (Unit 870), and activate for *contradiction* when the premise, but not the hypothesis, contains the word man. Others fire for verbs that are often associated with a specific label, such as sitting, eating, or sleeping. Many of these words have high *pointwise mutual information* (PMI) with the class prediction; as noted by [14], the top two highest words by PMI with *contradiction* are sleeping (15) and nobody (39, Figure 9). Still others (99) fire when there is high lexical overlap between premise and hypothesis, another heuristic in the literature [25]. Finally, there are neurons that are not well explained by this feature set (473). In general, we have found that many of the simple heuristics [14, 25] that make NLI models brittle to out-of-distribution data [13, 22, 37] are actually reified as individual features in deep representations.

## 5 Do interpretable neurons contribute to model accuracy?

A natural question to ask is whether it is empirically desirable to have more (or less) interpretable neurons, with respect to the kinds of concepts identified above. To answer this, we measure the performance of the entire model on the task of interest when the neuron is activated. In other words, for neuron $n$, what is the model accuracy on predictions for inputs where $M_n(\mathbf{x}) = 1$? In **image classification**, we find that the more interpretable the neuron (by IoU), the more accurate the model is when the neuron is active (Figure 7, left; $r = 0.31$, $p < 1e - 13$); the correlation increases as the formula length increases and we are better able to explain neuron behavior. Given that we are measuring abstractions over the human-annotated features deemed relevant for scene classification, this suggests, perhaps unsurprisingly, that neurons that detect more interpretable concepts are more accurate.

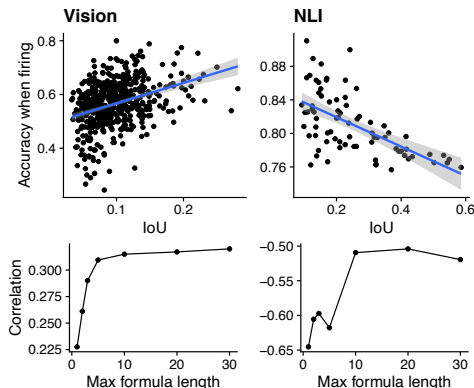

Figure 7: Top: neuron IoU versus model accuracy over inputs where the neuron is active for vision (length 10) and NLI (length 3). Bottom: Pearson correlation between these quantities versus max formula length.

However, when we apply the same analysis to the **NLI** model, the *opposite* trend occurs: neurons that we are better able to explain are *less* accurate (Figure 7, right; $r = -0.60$, $p < 1e-08$). Unlike vision, most sentence-level logical descriptions recoverable by our approach are spurious by definition, as they are too simple compared to the true reasoning required for NLI. If a neuron can be accurately summarized by simple deterministic rules, this suggests the neuron is making decisions based on spurious correlations, which is reflected by the lower performance. Analogously, the more *restricted* our feature set (by maximum formula length), the better we capture this anticorrelation. One important takeaway is that the "interpretability" of these explanations is not *a priori* correlated with performance, but rather dependent on the concepts we are searching for: given the right concept space, our method can identify behaviors that may be correlated *or* anticorrelated with task performance.

## 6 Can we target explanations to change model behavior?

Finally, we see whether compositional explanations allow us to manipulate model behavior. In both models, we have probed the final hidden representation before a final softmax layer produces the class predictions. Thus, we can measure a neuron's contribution to a specific class with the weight between the neuron and the class, and see whether constructing examples that activate (or inhibit) these neurons leads to corresponding changes in predictions. We call these "copy-paste" adversarial examples to differentiate them from standard adversarial examples involving imperceptible perturbations [36].

**Image Classification.** Figure 8 shows some Places365 classes along with the neurons that most contribute to the class as measured by the connection weight. In many cases, these connections are

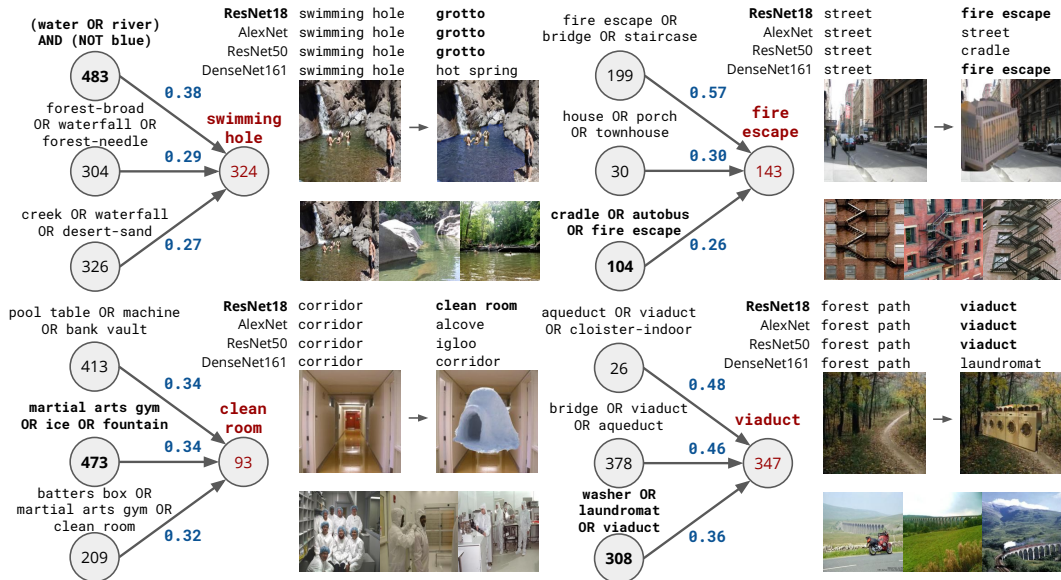

Figure 8: "copy-paste" adversarial examples for vision. For each **scene** (with 3 example images at bottom), the neuron that contribute most (by **connection weight**) are shown, along with their length 3 explanations. We target the **bold** explanations to crudely modify an input image and change the prediction towards/away from the scene. In the top-right corner, the left-most image is presented to the model (with predictions from 4 models shown); we modify the image to the right-most image, which changes the model prediction(s).

**Unit 39** (nobody in hypothesis)

```
hyp:nobody AND (NOT pre:hair) AND (NOT
pre:RB) AND (NOT pre:'s)
```
IoU **0.465**    $w_{entail}$ -0.117    $w_{neutral}$ -0.053    $w_{contra}$ 0.047

**Pre**      Three women prepare a meal in a kitchen.
**Orig Hyp** The ladies are cooking.
**Adv Hyp**  ***Nobody but*** the ladies are cooking.
True **entail** $\xrightarrow{adv}$ **neutral**    Pred **entail** $\xrightarrow{adv}$ **contra**

**Unit 15** (sitting only in hypothesis)

```
hyp:eating OR hyp:sitting OR hyp:sleeping OR
hyp:sits AND (NOT pre:sits)
```
IoU **0.239**    $w_{entail}$ -0.083 $w_{neutral}$ -0.059    $w_{contra}$ 0.086

**Orig Pre**  A blond woman is holding 2 golf balls while
              reaching down into a golf hole.
**Adv Pre**   A blond woman is holding 2 golf balls.
**Hyp**       A blond woman is sitting down.
True **contra** $\xrightarrow{adv}$ **neutral**  Pred **contra** $\xrightarrow{adv}$ **contra**

**Unit 133** (couch words in hypothesis)

```
NEIGHBORS(hyp:couch) OR hyp:inside OR
hyp:home OR hyp:indoors OR hype:eating
```
IoU **0.202**    $w_{entail}$ -0.125  $w_{neutral}$ -0.024    $w_{contra}$ 0.088

**Pre**      5 women sit around a table doing some crafts.
**Orig Hyp** 5 women sit around a table.
**Adv Hyp**  5 women sit around a table ***near a couch***.
True **entail** $\xrightarrow{adv}$ **neutral**    Pred **entail** $\xrightarrow{adv}$ **contra**

**Unit 941** (inside/indoors in hypothesis)

```
hyp:inside OR hyp:not OR hyp:indoors OR
hyp:moving OR hyp:something
```
IoU **0.151**    $w_{entail}$ 0.086  $w_{neutral}$ -0.030    $w_{contra}$ -0.023

**Orig Pre** Two people are sitting in a station.
**Adv Pre**  Two people are sitting in a ***pool***.
**Hyp**      A couple of people are inside and not standing.
True **entail** $\xrightarrow{adv}$ **neutral**    Pred **entail** $\xrightarrow{adv}$ **entail**

Figure 9: "copy-paste" adversarial examples for NLI. Taking an example from SNLI, we construct an **adversarial (adv)** premise or hypothesis which changes the true label and results in an *incorrect* model prediction (original label/prediction $\xrightarrow{adv}$ adversarial label/prediction).

sensible; water, foliage, and rivers contribute to a *swimming hole* prediction; houses, staircases, and fire escape (objects) contribute to *fire escape* (scene). However, the explanations in **bold** involve polysemanticity or spurious correlations. In these cases, we found it is possible to construct a "copy-paste" example which uses the neuron explanation to predictably alter the prediction.[2] In some cases, these adversarial examples are generalizable across networks besides the probed ResNet-18, causing the same behavior across AlexNet [24], ResNet-50 [15], and DenseNet-161 [21], all trained on Places365. For example, one major contributor to the *swimming hole* scene (top-left) is a neuron that fires for non-blue water; making the water blue switches the prediction to *grotto* in many models. The consistency of this misclassification suggests that models are detecting underlying biases in the

training data. Other examples include a neuron contributing to *clean room* that also detects ice and igloos; putting an igloo in a corridor causes a prediction to shift from *corridor* to *clean room*, though this does not occur across models, suggesting that this is an artifact specific to this model.

**NLI.** In NLI, we are able to trigger similar behavior by targeting spurious neurons (Figure 9). Unit 39 (top-left) detects the presence of `nobody` in the hypothesis as being highly indicative of *contradiction*. When creating an adversarial example by adding *nobody* to the original hypothesis, the true label shifts from *entailment* to *neutral*, but the model predicts *contradiction*. Other neurons predict *contradiction* when `couch`-related words (Unit 133) or `sitting` (Unit 15) appear in the hypothesis, and can be similarly targeted.

Overall, these examples are reminiscent of the image-patch attacks of [9], adversarial NLI inputs [1, 37], and the data collection process for recent *counterfactual* NLI datasets [13, 22], but instead of searching among neuron visualizations or using black-box optimization to synthesize examples, we use explanations as a transparent guide for crafting perturbations by hand.

# 7  Related Work

**Interpretability.** Interpretability in deep neural networks has received considerable attention over the past few years. Our work extends existing work on generating explanations for individual neurons in deep representations [4, 5, 10–12, 27], in contrast to analysis or probing methods that operate at the level of entire representations (e.g. [2, 19, 29]). Neuron-level explanations are fundamentally limited, since they cannot detect concepts distributed across multiple neurons, but this has advantages: first, neuron-aligned concepts offer evidence for representations that are *disentangled* with respect to concepts of interest; second, they inspect unmodified "surface-level" neuron behavior, avoiding recent debates on how complex representation-level probing methods should be [18, 29].

**Complex explanations.** In generating logical explanations of model behavior, one related work is the Anchors procedure of [33], which finds conjunctions of features that "anchor" a model's prediction in some local neighborhood in input space. Unlike Anchors, we do not explain local model behavior, but rather globally consistent behavior of neurons across an entire dataset. Additionally, we use not just conjunctions, but more complex compositions tailored to the domain of interest.

As our compositional formulas increase in complexity, they begin to resemble approaches to generating *natural language* explanations of model decisions [2, 8, 16, 17, 31]. These methods primarily operate at the representation level, or describe rationales for individual model predictions. One advantage of our logical explanations is that they are directly grounded in features of the data and have explicit measures of quality (i.e. IoU), in contrast to language explanations generated from black-box models that themselves can be uninterpretable and error-prone: for example, [17] note that naive language explanation methods often mention evidence not directly present in the input.

**Dataset biases and adversarial examples.** Complex neural models are often *brittle*: they fail to generalize to out-of-domain data [3, 13, 22, 32] and are susceptible to adversarial attacks where inputs are subtly modified in a way that causes a model to fail catastrophically [34, 36, 37]. This may be due in part to biases in dataset collection [3, 14, 30, 32], and models fail on datasets which eliminate these biases [3, 13, 22, 32]. In this work we suggest that these artifacts are learned to the degree that we can identify specific detectors for spurious features in representation space, enabling "copy-paste" adversarial examples constructed solely based on the explanations of individual neurons.

# 8  Discussion

We have described a procedure for obtaining compositional explanations of neurons in deep representations. These explanations more precisely characterize the behavior learned by neurons, as shown through higher measures of explanation quality (i.e. IoU) and qualitative examples of models learning perceptual abstractions in vision and spurious correlations in NLI. Specifically, these explanations (1) identify abstractions, polysemanticity, and spurious correlations localized to specific units in the representation space of deep models; (2) can disambiguate higher versus lower quality neurons in a model with respect to downstream performance; and (3) can be targeted to create "copy-paste" adversarial examples that predictably modify model behavior.

Several unanswered questions emerge:

1. We have limited our analysis in this paper to neurons in the penultimate hidden layers of our networks. Can we probe other layers, and better understand how concepts are formed and composed between the intermediate layers of a network (cf. [27])?

2. Does *model pruning* [20] more selectively remove the "lower quality" neurons identified by this work?

3. To what extent can the programs implied by our explanations serve as drop-in approximations of neurons, thus obviating the need for feature extraction in earlier parts of the network? Specifically, can we distill a deep model into a simple classifier over binary concept detectors defined by our neuron explanations?

4. If there is a relationship between neuron interpretability and model accuracy, as Section 5 has suggested, can we use neuron interpretability as a regularization signal during training, and does encouraging neurons to learn more interpretable abstractions result in better downstream task performance?

## Reproducibility

Code and data are available at `github.com/jayelm/compexp`.

## Broader Impact

Tools for model introspection and interpretation are crucial for better understanding the behavior of black-box models, especially as they make increasingly important decisions in high-stakes societal applications. We believe that the explanations generated in this paper can help unveil richer concepts that represent spurious correlations and potentially problematic biases in models, thus helping practitioners better understand the decisions made by machine learning models.

Nonetheless, we see two limitations with this method as it stands: (1) it currently requires technical expertise to implement, limiting usability by non-experts; (2) it relies on annotated datasets which may be expensive to collect, and may be biased in the kinds of features they contain (or omit). If a potential feature of interest is not present in the annotated dataset, it cannot appear in an explanation. Both of these issues can be ameliorated with future work in (1) building easier user interfaces for explainability, and (2) reducing data annotation requirements.

In high stakes cases, e.g. identifying model biases, care should also be taken to avoid relying too heavily on these explanations as causal proof that a model is encoding a concept, or assuming that the absence of an explanation is proof that the model does not encode the concept (or bias). We provide evidence that neurons exhibit surface-level behavior that is well-correlated with human-interpretable concepts, but by themselves, neuron-level explanations cannot identify the full array of concepts encoded in representations, nor establish definitive causal chains between inputs and decisions.

## Acknowledgments and Disclosure of Funding

Thanks to David Bau, Alex Tamkin, Mike Wu, Eric Chu, and Noah Goodman for helpful comments and discussions, and to anonymous reviewers for useful feedback. This work was partially supported by a gift from NVIDIA under the NVAIL grant program. JM is supported by an NSF Graduate Research Fellowship and the Office of Naval Research Grant ONR MURI N00014-16-1-2007.

## Footnotes

[1] https://spacy.io/

[2]Appendix B tests sensitivity of these examples to size and position of the copy-pasted subimages.

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
