[Supplementary Material]

# A  Concept uniqueness and granularity

Here, we report statistics about the uniqueness of neuron concepts, as we increase the maximum formula length of our explanations.

Figure S1: Number of repeated concepts across probed vision and NLI models, by maximum formula length.

Table S1: For probed **Image Classification** and **NLI** models, average number of occurrences of each detected concept and percentage of detected concepts that are unique (i.e. appear only once).

|  | **Image Classification** | | **NLI** | |
| $N$ | Mean concept count | % unique | Mean concept count | % unique |
| --- | --- | --- | --- | --- |
| 1 | 2.61 | 42% | 3.30 | 39% |
| 3 | 1.03 | 97% | 1.20 | 86% |
| 5 | 1.01 | 99% | 1.04 | 96% |
| 10 | 1.00 | 100% | 1.00 | 100% |

## A.1  Image Classification

Figure S1 (left) plots the number of times each unique concept appears across the 512 units of ResNet-18 as the maximum formula length increases. Table S1 displays the mean number of occurrences per concept, and percentage of concepts occurring that are unique (i.e. occur only once). At length 1 (equivalent to NetDissect), many concepts appear multiple times, where only 42% of concepts occur only once and the mean number of occurrences is 2.61. But uniqueness increases dramatically as the formula length increases: already by length 3, 97% of concepts are unique, and concepts are all unique by length 10. Our explanations thus reveal significant specialization in neuron function (vs. NetDissect [5]). Table S2 shows the most common repeated concepts for each maximum formula lengths.

## A.2  NLI

Similarly, Figure S1 (right) plots the number of times each unique concept appears across the neurons of the NLI model, and Table S1 displays occurrence statistics. Like the Image Classification model, longer formula lengths reveal significantly more specialization in neuron function. Table S3 shows the most common repeated concepts for each maximum formula lengths.

# B  Adversarial example sensitivity

In Figure S2 we vary the size and position of subimages for the copy-paste examples (note this analysis is less straightforward for examples like *non-blue water*). Sensitivity depends on the specific example. In general, if the sub-image is too small (left), the original class prevails; otherwise, the *igloo → clean room* example is quite reliable, while the *street → fire escape* example is less so.

Table S2: Up to the 10 most repeated concepts in ResNet-18 conv4 by length $N$. At $N = 5$ there is only 1 non-unique concept. For a full list see the code.

| $N$ | Concept | # |
|---|---|---|
| 1 | pool table | 15 |
| | house | 12 |
| | corridor | 11 |
| | cockpit | 10 |
| | bed | 10 |
| | bakery shop | 9 |
| | bathroom | 9 |
| | alley | 8 |
| | airport terminal | 8 |
| | car | 8 |
| 3 | pillow OR (bed AND bedroom) | 4 |
| | sink OR toilet OR bathtub | 3 |
| | auditorium OR movie theater OR conference center | 2 |
| | sink OR toilet OR countertop | 2 |
| | pool table OR arcade machine OR slot machine | 2 |
| | pool table OR golf course OR fairway | 2 |
| | greenhouse OR vegetable garden OR herb garden | 2 |
| | water OR river AND (NOT blue) | 2 |
| | living room AND (sofa OR cushion) | 2 |
| | street AND sky AND white | 2 |
| 5 | auditorium OR indoor theater OR conference center OR movie theater OR silver screen | 2 |

Table S3: Up to 10 of the most repeated concepts in our probed NLI baseline model by length $N$. At $N = 3$ there are only 9 non-unique concepts; at $N = 5$ there are only 3 non-unique concepts. For a full list see the code.

| | NLI | |
|---|---|---|
| $N$ | Concept | Count |
| 1 | pre:NN | 17 |
| | hyp:NN | 14 |
| | hyp:IN | 8 |
| | overlap-75% | 4 |
| | hyp:VBG | 4 |
| | hyp:in | 3 |
| | hyp:. | 3 |
| | hyp:sitting | 2 |
| | hyp:DT | 2 |
| | hyp:for | 2 |
| 3 | pre:NN AND (NOT overlap-50%) AND (not hyp:outside) | 4 |
| | pre:NN AND (NOT hyp:for) AND (NOT hyp:VB) | 3 |
| | (NOT hyp:sleeping) AND (pre:NN OR hyp:NNS) | 3 |
| | hyp:NN AND (NOT overlap-50%) AND (NOT hyp:outside) | 2 |
| | hyp:for OR hyp:PRP$ OR hyp:to | 2 |
| | hyp:NN AND (NOT overlap-50%) AND (NOT hyp:there) | 2 |
| | pre:NN AND (NOT overlap-75%) AND (NOT hyp:outside) | 2 |
| | pre:NN AND (NOT hyp:for) AND (NOT hyp:PRP$) | 2 |
| | hyp:eating OR (hyp:IN AND (NOT overlap-50%)) | 2 |
| 5 | hyp:NNP OR ((NOT hyp:EX) AND (hyp:IN OR hyp:PRP$ OR hyp:to)) | 2 |
| | (NOT hyp:for) AND (NOT hyp:PRP) AND (overlap-50% OR (pre:NN AND (NOT hyp:PRP$))) | 2 |
| | pre:NN AND (NOT overlap-50%) AND (NOT hyp:outside) AND (NOT hyp:EX) AND (NOT hyp:outdoors) | 2 |

(a) street + cradle = fire escape

(b) corridor + igloo = clean room

(c) forest path + laundry machines = viaduct

Figure S2: Varying the size and position of pasted sub-images for the vision copy-paste adversarial examples. Green: prediction changes to intended adversarial class; yellow: prediction changes to a different class (e.g. *aqueduct* for the bottom row); red = no change.

# C  Additional image classification examples

Examples are not cherry picked; we enumerate neurons 0–39.

**Unit 0** (lexical and perceptual: bars/surfaces)

**Length** 1  **IoU** 0.075  reception

**Length** 3  **IoU** 0.106  ((reception OR work surface) OR bar)

**Length** 10 **IoU** 0.107  (((((((((reception OR work surface) OR bar) OR ticket counter) OR button panel) AND (NOT desk)) AND (NOT table)) AND (NOT home office)) AND (NOT bottle)) AND (NOT drawer))

**Unit 1** (lexical and perceptual: windows/shelves)

**Length** 1  **IoU** 0.045  shop window

**Length** 3  **IoU** 0.084  ((shop window OR pantry) OR liquor store outdoor)

**Length** 10 **IoU** 0.103  (((((((((shop window OR pantry) OR liquor store outdoor) OR shopfront) OR toyshop) OR gift shop) OR convenience store outdoor) OR pub outdoor) OR trade name) OR shoe shop)

**Unit 2** (lexical and perceptual: islands/grass/water)

**Length** 1  **IoU** 0.063  sea

**Length** 3  **IoU** 0.078  ((sea OR marsh) OR lake)

**Length** 10 **IoU** 0.090  (((((((((sea OR marsh) OR lake) OR island) AND (NOT ocean)) OR bayou) OR golf course) AND (NOT tree)) OR bog) OR watering hole)

**Unit 3** (lexical and perceptual: screens)

**Length** 1  **IoU** 0.094  auditorium

**Length** 3  **IoU** 0.177  ((auditorium OR movie theater indoor) OR theater indoor procenium)

**Length** 10 **IoU** 0.230  (((((((((auditorium OR movie theater indoor) OR theater indoor procenium) OR silver screen) OR conference center) AND (NOT ceiling)) OR blackboard) AND (NOT floor)) OR lecture room) AND (NOT ceiling))

**Unit 4** (polysemantic: columns/chandeliers)

**Length** 1  **IoU** 0.025  courthouse

**Length** 3  **IoU** 0.049  ((courthouse OR throne room) OR ballroom)

**Length** 10 **IoU** 0.064  (((((((((courthouse OR throne room) OR chandelier) OR column) OR ballroom) OR bandstand) OR embassy) AND (NOT living room)) AND (NOT poolroom home)) AND (NOT courtroom))

**Unit 5** (perceptual only: debris)

**Length** 1  **IoU** 0.033  shelf

**Length** 3  **IoU** 0.062  ((slum OR toilet) OR pantry)

**Length** 10 **IoU** 0.074  (((((((((slum OR toilet) OR pantry) OR rubbish) OR bazaar outdoor) OR bucket) OR washer) OR plaything) OR workshop) AND (NOT market outdoor))

**Unit 6** (lexical and perceptual: domes)

**Length** 1  **IoU** 0.049  gazebo exterior

**Length** 3  **IoU** 0.073  ((gazebo exterior OR tent) OR dome)

**Length** 10 **IoU** 0.098  (((((((((gazebo exterior OR tent) OR dome) OR big top) OR hut) OR island) OR butte) OR mausoleum) OR greenhouse) OR bandstand)

**Unit 7** (lexical and perceptual: fields of plants)

**Length** 1  **IoU** 0.057  vineyard

**Length** 3  **IoU** 0.127  ((vineyard OR orchard) OR corn field)

**Length** 10 **IoU** 0.133  ((((vineyard OR orchard) OR corn field) OR vineyard) AND (NOT hedge))

**Unit 8** (lexical and perceptual: plants)

**Length** 1  **IoU** 0.098  greenhouse indoor

**Length** 3  **IoU** 0.130  ((greenhouse indoor OR greenhouse) OR vegetable garden)

**Length** 10  **IoU** 0.156  (((((((((greenhouse indoor OR greenhouse) OR vegetable garden) OR vineyard) OR leaves) OR florist shop indoor) OR corn field) OR leaf) AND (NOT field)) OR vegetable garden)

**Unit 9** (polysemantic: grass/balls/other)

**Length** 1  **IoU** 0.037  grass

**Length** 3  **IoU** 0.050  ((ball pit OR plaything) OR kindergarden classroom)

**Length** 10  **IoU** 0.053  (((((((((ball pit OR plaything) OR kindergarden classroom) OR fruit) OR day care center) OR lake artificial) AND (NOT water)) AND (NOT painting)) AND (NOT pedestal)) AND (NOT fence))

**Unit 10** (polysemantic: mountains/highway)

**Length** 1  **IoU** 0.065  highway

**Length** 3  **IoU** 0.088  ((highway OR field cultivated) AND (NOT sky))

**Length** 10  **IoU** 0.092  (((((((((highway OR field cultivated) AND (NOT sky)) OR wheat field) OR desertand) AND (NOT sky)) OR field road) AND (NOT sky)) OR mountain road) AND (NOT earth))

**Unit 11** (polysemantic: people/others)

**Length** 1  **IoU** 0.072  person

**Length** 3  **IoU** 0.077  ((person OR booth indoor) AND (NOT art studio))

**Length** 10  **IoU** 0.078  (((((((((person AND (NOT pink)) OR booth indoor) AND (NOT art studio)) AND (NOT head)) OR market indoor) OR booth indoor) OR torso) AND (NOT conference center)) AND (NOT fire station))

**Unit 12** (polysemantic: dining rooms/fire escapes/others)

**Length** 1  **IoU** 0.042  dining room

**Length** 3  **IoU** 0.072  ((dining room AND table) OR fire escape)

**Length** 10  **IoU** 0.085  (((((((((dining room AND table) OR fire escape) OR catacomb) OR shelter) OR throne room) OR altar) OR altarpiece) OR fire escape) AND (NOT chair))

**Unit 13** (polysemantic: islands/canopies/others)

**Length** 1  **IoU** 0.035  islet

**Length** 3  **IoU** 0.070  ((islet OR cavern indoor) OR canopy)

**Length** 10  **IoU** 0.089  (((((((((islet OR cavern indoor) OR canopy) OR rope bridge) OR carousel) OR catacomb) OR bedchamber) OR altarpiece) OR niche) OR bayou)

**Unit 14** (perceptual only: fences/horizontal lines)

**Length** 1  **IoU** 0.038  boxing ring

**Length** 3  **IoU** 0.061  ((boxing ring OR corral) OR bleachers indoor)

**Length** 10  **IoU** 0.081  (((((((((boxing ring OR corral) OR bleachers indoor) OR fence) OR military hut) OR wrestling ring indoor) OR bakery kitchen) OR barnyard) OR parking garage outdoor) OR horse)

**Unit 15** (perceptual only: beds)

**Length** 1  **IoU** 0.062  operating room

**Length** 3  **IoU** 0.104  ((operating room OR hospital room) OR cradle)

**Length** 10  **IoU** 0.105  (((((((((operating room OR hospital room) OR cradle) OR dentists office) AND (NOT sink)) AND (NOT chest of drawers)) OR operating room) AND (NOT drawer)) AND (NOT footboard))

**Unit 16** (polysemantic: gyms/windmills/other)

**Length** 1   **IoU** 0.031   `ice skating rink indoor`

**Length** 3   **IoU** 0.064   `((ice skating rink indoor OR basketball court indoor) OR martial arts gym)`

**Length** 10   **IoU** 0.112   `(((((((((ice skating rink indoor OR basketball court indoor) OR martial arts gym) OR windmill) OR hangar indoor) OR boxing ring) OR wrestling ring indoor) OR fire escape) OR badminton court indoor) OR subway station corridor)`

**Unit 17** (perceptual only: flat areas)

**Length** 1   **IoU** 0.068   `auditorium`

**Length** 3   **IoU** 0.101   `((auditorium OR conference center) OR movie theater indoor)`

**Length** 10   **IoU** 0.112   `(((((((((auditorium OR conference center) OR movie theater indoor) OR theater indoor procenium) OR silver screen) OR courtroom) AND (NOT bench)) AND (NOT pedestal)) OR auditorium) AND (NOT swivel chair))`

**Unit 18** (polysemantic: rocks/forests/other)

**Length** 1   **IoU** 0.047   `badlands`

**Length** 3   **IoU** 0.072   `((badlands OR forest needleleaf) OR slot machine)`

**Length** 10   **IoU** 0.081   `(((((((((badlands OR forest needleleaf) OR slot machine) OR junkyard) OR arcade machine) OR cow) OR semidesert ground) OR animal) OR car interior backseat) AND (NOT green))`

**Unit 19** (lexical and perceptual: cases)

**Length** 1   **IoU** 0.144   `bakeryhop`

**Length** 3   **IoU** 0.173   `((bakeryhop OR case) AND (NOT supermarket))`

**Length** 10   **IoU** 0.188   `(((((((((bakeryhop OR case) OR food) AND (NOT supermarket)) OR bakery kitchen) OR butchers shop) OR ice cream parlor) OR island) AND (NOT kitchen)) AND (NOT cabinet))`

**Unit 20** (lexical and perceptual: houses/decks)

**Length** 1   **IoU** 0.036   `house`

**Length** 3   **IoU** 0.044   `((house OR motel) OR zen garden)`

**Length** 10   **IoU** 0.049   `(((((((((house OR motel) OR zen garden) OR hunting lodge outdoor) OR lido deck outdoor) AND (NOT house)) OR student residence) OR swimming pool indoor) OR barnyard) AND (NOT barn))`

**Unit 21** (polysemantic: bookcases/fire stations)

**Length** 1   **IoU** 0.063   `bookcase`

**Length** 3   **IoU** 0.098   `((bookcase OR fire station) OR book)`

**Length** 10   **IoU** 0.116   `(((((((((bookcase OR fire station) OR book) OR videostore) OR garage door) OR library indoor) AND (NOT archive)) OR videos) OR convenience store indoor) OR exhibitor)`

**Unit 22** (lexical and perceptual: bridges, possibly over water)

**Length** 1   **IoU** 0.035   `river`

**Length** 3   **IoU** 0.066   `((river OR bridge) OR rope bridge)`

**Length** 10   **IoU** 0.080   `(((((((((river OR bridge) OR rope bridge) OR creek) OR mountain path) OR aqueduct) OR gulch) OR sandbar) OR footbridge) AND (NOT canal natural))`

**Unit 23** (perceptual only: vertical/perspective lines)

**Length** 1   **IoU** 0.046   `kitchen`

**Length** 3   **IoU** 0.058   `((youth hostel OR stove) OR galley)`

**Length** 10   **IoU** 0.070   `(((((((((youth hostel OR stove) OR galley) OR microwave) OR work surface) OR telephone booth) OR cubicle office) OR kitchenette) OR exhaust hood) AND (NOT drawer))`

**Unit 24** (polysemantic: beds/fireplaces/other)

**Length** 1  **IoU** 0.053  fireplace

**Length** 3  **IoU** 0.058  ((fireplace OR buffet) OR pulpit)

**Length** 10 **IoU** 0.060  ((((((((((fireplace OR buffet) OR pulpit) OR microwave) AND (NOT poolroom home)) AND (NOT wet bar)) AND (NOT pane)) AND (NOT dinette home)) AND (NOT church indoor)) OR microwave)

**Unit 25** (perceptual only: empty corridors)

**Length** 1  **IoU** 0.067  corridor

**Length** 3  **IoU** 0.083  ((corridor OR sauna) OR elevator)

**Length** 10 **IoU** 0.087  (((((((((corridor OR sauna) OR elevator) OR basement) OR fire escape) OR elevator door) OR cargo container interior) OR elevator freight elevator) AND (NOT door frame)) OR corridor)

**Unit 26** (lexical and perceptual: aqueducts)

**Length** 1  **IoU** 0.042  aqueduct

**Length** 3  **IoU** 0.079  ((aqueduct OR viaduct) OR cloister indoor)

**Length** 10 **IoU** 0.097  (((((((((aqueduct OR viaduct) OR cloister indoor) OR bandstand) OR arch) OR aqueduct) OR viaduct) OR water tower) OR arcade) OR arcades)

**Unit 27** (perceptual only: dome-like things)

**Length** 1  **IoU** 0.032  cockpit

**Length** 3  **IoU** 0.054  ((cockpit OR wave) OR viaduct)

**Length** 10 **IoU** 0.066  (((((((((cockpit OR wave) OR viaduct) OR hovel) OR tent) OR dam) OR fountain) OR ice) OR dolmen) OR viaduct)

**Unit 28** (lexical and perceptual: mediterranean houses)

**Length** 1  **IoU** 0.045  alley

**Length** 3  **IoU** 0.081  ((medina OR kasbah) OR alley)

**Length** 10 **IoU** 0.092  (((((((medina OR kasbah) OR alley) AND building) OR kasbah) OR medina) AND (NOT railing))

**Unit 29** (lexical and perceptual: house facades)

**Length** 1  **IoU** 0.088  house

**Length** 3  **IoU** 0.092  ((house OR porch) OR town house)

**Length** 10 **IoU** 0.093  (((((((((house AND (NOT building facade)) OR porch) OR town house) OR inn outdoor) AND (NOT plant)) AND (NOT alley)) AND (NOT dacha)) AND (NOT stairs)) AND (NOT general store outdoor))

**Unit 30** (lexical and perceptual: porches)

**Length** 1  **IoU** 0.075  balcony interior

**Length** 3  **IoU** 0.088  ((balcony interior OR dinette home) OR control tower indoor)

**Length** 10 **IoU** 0.089  (((((balcony interior OR dinette home) OR control tower indoor) AND (NOT door)) AND (NOT curtain)) AND (NOT armchair))

**Unit 31** (polysemantic: pool tables/others)

**Length** 1  **IoU** 0.106  pool table

**Length** 3  **IoU** 0.124  ((pool table OR arcade machine) OR television camera)

**Length** 10 **IoU** 0.126  (((((pool table OR arcade machine) OR television camera) OR table tennis) AND (NOT television studio)) AND (NOT wet bar)) AND (NOT music studio))

**Unit 32** (perceptual only: red things)

**Length** 1    **IoU** 0.045    `red`

**Length** 3    **IoU** 0.058    `((fire station OR bullring) OR boxing ring)`

**Length** 10    **IoU** 0.069    `(((((((((fire station OR bullring) OR boxing ring) OR throne room) OR telephone booth) OR big top) OR ring) OR joss house) OR autobus) AND (NOT grandstand))`

**Unit 33** (lexical and perceptual: landscapes/horizons)

**Length** 1    **IoU** 0.078    `badlands`

**Length** 3    **IoU** 0.116    `((badlands OR desertand) OR oasis)`

**Length** 10    **IoU** 0.132    `((((((((((badlands OR desertand) OR oasis) OR hoodoo) OR bulldozer) OR canyon) OR dam) AND (NOT rock)) OR badlands) AND (NOT tree))`

**Unit 34** (polysemantic: beds and shelves)

**Length** 1    **IoU** 0.028    `bed`

**Length** 3    **IoU** 0.032    `((childs room OR dorm room) OR youth hostel)`

**Length** 10    **IoU** 0.034    `((((((((childs room OR dorm room) OR youth hostel) OR cushion) OR pantry) OR pillow) AND (NOT wardrobe)) AND (NOT door)) AND (NOT carpet)) AND (NOT attic))`

**Unit 35** (polysemantic: water/other structures)

**Length** 1    **IoU** 0.019    `beach`

**Length** 3    **IoU** 0.029    `((beach OR tent) OR caravan)`

**Length** 10    **IoU** 0.039    `((((((((beach OR tent) OR caravan) OR hovel) OR bayou) OR manufactured home) OR watering hole) OR oasis) OR excavation) OR junkyard)`

**Unit 36** (perceptual only: complex white structures)

**Length** 1    **IoU** 0.041    `boat`

**Length** 3    **IoU** 0.062    `((boat OR ship) OR aircraft carrier)`

**Length** 10    **IoU** 0.082    `((((((((((boat OR ship) OR aircraft carrier) OR lighthouse) OR cannon) OR workshop) OR pier) OR roller coaster) OR water tower) OR dam)`

**Unit 37** (perceptual only: empty halls/rooms)

**Length** 1    **IoU** 0.030    `corridor`

**Length** 3    **IoU** 0.049    `((airplane cabin OR subway interior) OR berth)`

**Length** 10    **IoU** 0.062    `((((((((((airplane cabin OR subway interior) OR berth) OR operating room) OR hospital room) OR gymnasium indoor) OR swivel chair) AND (NOT conference room)) OR pilothouse indoor) AND (NOT desk))`

**Unit 38** (perceptual only: things on grass)

**Length** 1    **IoU** 0.034    `lighthouse`

**Length** 3    **IoU** 0.060    `((lighthouse OR bullring) OR batters box)`

**Length** 10    **IoU** 0.076    `((((((((((lighthouse OR bullring) OR batters box) OR fairway) OR water tower) OR plane) OR pitch) OR baseball field) AND (NOT sky)) OR lighthouse)`

**Unit 39** (perceptual only: flat surfaces)

**Length** 1    **IoU** 0.038    `bed`

**Length** 3    **IoU** 0.048    `((pool table OR pillow) OR swimming pool)`

**Length** 10    **IoU** 0.054    `(((((((((pool table OR pillow) OR swimming pool) OR cushion) OR hotel outdoor) AND (NOT black)) AND (NOT swimming pool indoor)) OR eiderdown) AND (NOT black)) OR pillow)`

# D   Additional NLI examples

Examples are not cherry picked; we enumerate the first 25 neurons that fire reliably (i.e. at least 500 times across the validation dataset), skipping those already illustrated in the main paper.

**Unit 0**

`(((((NOT overlap-50%) AND pre:NN) AND (NOT hyp:VB)) AND (NOT hyp:outside)) AND (NOT hyp:near))`

IoU **0.355**  $w_{entail}$ -0.027   $w_{neutral}$ -0.018   $w_{contra}$ 0.027

| | |
|---|---|
| **Pre** | a woman dressed in a blue long - sleeved shirt and wearing a hairnet . |
| **Hyp** | the woman is naked and alone in the bathroom . |

Act **43.58**  True **contra**  Pred **contra**

| | |
|---|---|
| **Pre** | two men are on a cherry picker proceeding to perform work at a construction site . |
| **Hyp** | two men driving in a truck down an empty highway . |

Act **42.50**  True **contra**  Pred **contra**

| | |
|---|---|
| **Pre** | these two poodles , one black and one brown , are playing . |
| **Hyp** | the cats are brown and red . |

Act **41.24**  True **contra**  Pred **contra**

---

**Unit 6**

`(((((NOT overlap-25%) AND pre:NN) AND (NOT hyp:people)) AND (NOT hyp:EX)) OR hyp:tall)`

IoU **0.239**  $w_{entail}$ -0.063   $w_{neutral}$ 0.022   $w_{contra}$ 0.009

| | |
|---|---|
| **Pre** | a man in a blue helmet jumping off of a hill on a dirt bike . |
| **Hyp** | the man is a professional athlete . |

Act **26.31**  True **neutral**  Pred **neutral**

| | |
|---|---|
| **Pre** | a man standing in front of a class of asian students holding a picture of santa claus . |
| **Hyp** | a tall human standing |

Act **26.02**  True **neutral**  Pred **neutral**

| | |
|---|---|
| **Pre** | a girl prepares herself for the swim meet . |
| **Hyp** | the girl has swam before . |

Act **25.25**  True **entail**  Pred **neutral**

---

**Unit 8**

`(((((hyp:for OR hyp:to) OR hyp:tall) OR hyp:their) AND (NOT hyp:next))`

IoU **0.247**  $w_{entail}$ -0.015   $w_{neutral}$ 0.023   $w_{contra}$ 0.000

| | |
|---|---|
| **Pre** | a man is doing tricks on a skateboard . |
| **Hyp** | a tall human doing tricks |

Act **29.89**  True **neutral**  Pred **neutral**

| | |
|---|---|
| **Pre** | a guy on inline skates with a white hat is on a yellow rail . |
| **Hyp** | the guy on inline skates is trying to impress his girlfriend . |

Act **26.10**  True **neutral**  Pred **neutral**

| | |
|---|---|
| **Pre** | a gentleman in a striped shirt gesturing with a stick - like object in his hand while passersby stare at him . |
| **Hyp** | a gentleman in a striped shirt joyously gesturing |

Act **24.58**  True **neutral**  Pred **neutral**

---

**Unit 16**

`(((((NOT hyp:wearing) AND pre:NN) AND (NOT hyp:sleeping)) AND (NOT hyp:sitting)) AND (NOT hyp:eating))`

IoU **0.387**  $w_{entail}$ 0.022   $w_{neutral}$ 0.010   $w_{contra}$ -0.042

| | |
|---|---|
| **Pre** | a woman wearing a red scarf raises her hand as she walks in a parade . |
| **Hyp** | a woman raises her hand as she walks in a parade for st. patrick 's day . |

Act **32.96**  True **neutral**  Pred **neutral**

| | |
|---|---|
| **Pre** | a guy on inline skates with a white hat is on a yellow rail . |
| **Hyp** | the guy on inline skates is trying to impress his girlfriend . |

Act **29.88**  True **neutral**  Pred **neutral**

| | |
|---|---|
| **Pre** | three men ; one pedaling while playing drums , one playing piano and one both pedaling and steering , move a type of mobile band down a street . |
| **Hyp** | three men are trying to attract a crowd and take them to a bar where they will be playing later |

Act **28.13**  True **neutral**  Pred **neutral**

---

**Unit 70**

`(((((hyp:in OR hyp:nobody) OR hyp:sitting) AND (NOT overlap-75%)) OR hyp:cat)`

IoU **0.164**  $w_{entail}$ -0.095    $w_{neutral}$ -0.019    $w_{contra}$ 0.051

| | |
|---|---|
| **Pre** | many people have painted faces at night . |
| **Hyp** | the people are swimming in the ocean at noon . |
| Act **39.61** | True **contra**   Pred **contra** |
| **Pre** | a man is carrying a child while holding a red and blue umbrella . |
| **Hyp** | a man is swimming laps in a pool . |
| Act **39.46** | True **contra**   Pred **contra** |
| **Pre** | a man with a mustache is playing ice hockey with snow in the background . |
| **Hyp** | people are swimming in the lake . |
| Act **38.12** | True **contra**   Pred **contra** |

## Unit 71
`(((((NOT hyp:to) AND pre:NN) AND (NOT hyp:for)) AND (NOT overlap-75%)) AND (NOT hyp:outdoors))`
IoU **0.366**  $w_{entail}$ 0.005    $w_{neutral}$ -0.049    $w_{contra}$ 0.022

| | |
|---|---|
| **Pre** | a young man smiles and points at something off - camera , while standing in front of a display . |
| **Hyp** | the young man is frowning with his hands in his pockets . |
| Act **37.08** | True **contra**   Pred **contra** |
| **Pre** | a little boy in a blue shirt holding a toy . |
| **Hyp** | boy dressed in red lighting things on fire . |
| Act **36.44** | True **contra**   Pred **contra** |
| **Pre** | a shepherd breed dog running on the beach |
| **Hyp** | a dog is at home sleeping |
| Act **36.19** | True **contra**   Pred **contra** |

## Unit 89
`(((((NOT overlap-50%) AND pre:NN) AND (NOT pre:for)) AND (NOT hyp:sitting)) AND (NOT hyp:wearing))`
IoU **0.251**  $w_{entail}$ -0.054    $w_{neutral}$ 0.015    $w_{contra}$ 0.024

| | |
|---|---|
| **Pre** | a little girl with a hat sits between a woman 's feet in the sand in front of a pair of colorful tents . |
| **Hyp** | the girl is related to the woman . |
| Act **33.38** | True **neutral**   Pred **neutral** |
| **Pre** | two girls are sitting outside on the ground in front of a lake . |
| **Hyp** | two girls waiting for butterflies |
| Act **30.10** | True **neutral**   Pred **neutral** |
| **Pre** | three hockey players are in the middle of a play . |
| **Hyp** | the players are playing for the championship |
| Act **27.38** | True **neutral**   Pred **neutral** |

## Unit 98
`(((((NOT overlap-50%) AND (hyp:in OR hyp:running)) OR hyp:swimming) OR hyp:riding)`
IoU **0.127**  $w_{entail}$ -0.099    $w_{neutral}$ -0.035    $w_{contra}$ 0.061

| | |
|---|---|
| **Pre** | a woman , wearing a dress , while sitting down playing a musical instrument and singing into a microphone . |
| **Hyp** | the woman is swimming in the middle of the ocean by herself . |
| Act **35.66** | True **contra**   Pred **contra** |
| **Pre** | a man with a mustache is playing ice hockey with snow in the background . |
| **Hyp** | people are swimming in the lake . |
| Act **35.56** | True **contra**   Pred **contra** |
| **Pre** | people walking through dirt . |
| **Hyp** | people are swimming . |
| Act **32.38** | True **contra**   Pred **contra** |

## Unit 128
`(((((NOT overlap-50%) AND hyp:NN) AND (NOT hyp:outside)) OR hyp:sleeping) AND (NOT hyp:near))`
IoU **0.313**  $w_{entail}$ -0.035    $w_{neutral}$ 0.001    $w_{contra}$ 0.034

| | |
|---|---|
| **Pre** | two men are on a cherry picker proceeding to perform work at a construction site . |
| **Hyp** | two men driving in a truck down an empty highway . |
| Act **46.77** | True **contra**   Pred **contra** |
| **Pre** | a woman dressed in a blue long - sleeved shirt and wearing a hairnet . |

**Hyp**    the woman is naked and alone in the bathroom .
Act **44.54**   True **contra**   Pred **contra**
**Pre**    a boy in a red shirt and a boy in a yellow shirt are jumping on a trampoline outside .
**Hyp**    the boys are asleep .
Act **42.99**   True **contra**   Pred **contra**

---

## Unit 134

`(((((NOT pre:blue) AND (hyp:. AND hyp:NN)) AND (NOT hyp:there)) AND (NOT hyp:outside))`
IoU **0.200**   $w_{entail}$ -0.055      $w_{neutral}$ 0.009      $w_{contra}$ 0.038

**Pre**    a man in a red hat and shirt with gray shorts attempts to do the splits .
**Hyp**    the man has a blue hat .
Act **29.96**   True **contra**   Pred **contra**

**Pre**    people walking down a busy city street in the winter .
**Hyp**    people are running down a busy city street in summer .
Act **28.37**   True **contra**   Pred **contra**

**Pre**    police officer and his motorcycle in a crowd of people at a protest .
**Hyp**    a police officer is riding a unicorn in front of a crowd .
Act **27.77**   True **contra**   Pred **contra**

---

## Unit 157

`(((((hyp:for OR hyp:to) AND hyp:.) OR hyp:asleep) OR hyp:sad)`
IoU **0.150**   $w_{entail}$ -0.032      $w_{neutral}$ 0.026      $w_{contra}$ -0.032

**Pre**    a pale dog runs down a path .
**Hyp**    a dog is running towards his owner
Act **21.24**   True **neutral**   Pred **neutral**

**Pre**    a man in a black and blue jacket and a white helmet skiing down a hill swiftly .
**Hyp**    a man goes down the ski hill swiftly because he is an expert .
Act **20.73**   True **neutral**   Pred **neutral**

**Pre**    a woman wearing a red scarf raises her hand as she walks in a parade .
**Hyp**    a woman raises her hand as she walks in a parade for st. patrick 's day .
Act **20.34**   True **neutral**   Pred **neutral**

---

## Unit 173

`((((((NOT overlap-75%) AND hyp:IN) OR pre:sitting) OR pre:water) AND (NOT hyp:there))`
IoU **0.175**   $w_{entail}$ -0.085      $w_{neutral}$ -0.021      $w_{contra}$ 0.035

**Pre**    a mother and her two children sit down to rest .
**Hyp**    three people are running around .
Act **31.63**   True **contra**   Pred **contra**

**Pre**    a group of people sitting in a grassy area under a pink and white blossoming tree .
**Hyp**    people are running in a grassy area .
Act **30.69**   True **contra**   Pred **contra**

**Pre**    3 people sitting in a boat , rowing in a large body of water surrounded by greenery .
**Hyp**    the people standing in a train
Act **30.24**   True **contra**   Pred **contra**

---

## Unit 203

`(((((NOT overlap-50%) AND (hyp:in OR hyp:on)) OR hyp:sleeping) OR hyp:eating)`
IoU **0.167**   $w_{entail}$ -0.061      $w_{neutral}$ -0.009      $w_{contra}$ 0.059

**Pre**    a girl and two boys are playing in water .
**Hyp**    the children are eating dinner at a restaurant .
Act **37.10**   True **contra**   Pred **contra**

**Pre**    while some people look in the barn , others walk on the bridge and some are enjoying cooling off in the water by the beach .
**Hyp**    the people are going in the barn to see the horse .
Act **34.58**   True **neutral**   Pred **contra**

**Pre**    brown dog running through shallow water .
**Hyp**    a dog is sleeping on a blanket .
Act **33.11**   True **contra**   Pred **contra**

## Unit 257

`((((hyp:their OR overlap-75%) AND hyp:IN) OR hyp:friend) AND hyp:.)`

IoU **0.218**   w_entail -0.041      w_neutral 0.052      w_contra 0.003

**Pre**   a dressed up woman walking next to a store at night .
**Hyp**   a dressed up woman is walking next to a pharmacy at night .
Act **30.76**   True **neutral**   Pred **neutral**

**Pre**   a man in a blue shirt , khaki shorts , ball cap and white socks and loafers walking behind a group of people walking down a stone walkway with a water bottle in his left hand .
**Hyp**   a man in a blue shirt , khaki shorts , ball cap and blue socks and loafers walking behind a group of people walking down a stone walkway with a water bottle in his left hand .
Act **29.69**   True **contra**   Pred **contra**

**Pre**   a man is standing in coconuts while trying to open one .
**Hyp**   a sad man is standing in coconuts while trying to open one .
Act **29.24**   True **neutral**   Pred **neutral**

## Unit 265

`(((((NOT hyp:PRP$) AND pre:.) AND (NOT hyp:VB)) AND (NOT hyp:PRP)) AND (NOT hyp:in))`

IoU **0.323**   w_entail 0.028      w_neutral -0.003      w_contra -0.055

**Pre**   a youth is kicking a soccer ball in an empty brick area .
**Hyp**   a human kicking .
Act **35.21**   True **entail**   Pred **entail**

**Pre**   a band of people playing brass instruments is performing outside .
**Hyp**   a group of people have instruments .
Act **31.93**   True **entail**   Pred **entail**

**Pre**   three hikers are hiking in a mountain filled with trees and snow .
**Hyp**   peopl;e were on grass
Act **31.77**   True unknown   Pred **entail**

## Unit 270

`(((((NOT overlap-50%) AND hyp:DT) AND (NOT hyp:outside)) AND (NOT hyp:has)) AND (NOT hyp:near))`

IoU **0.315**   w_entail -0.060      w_neutral -0.011      w_contra 0.030

**Pre**   several people prepare their stalls that consist of fish , vegetables and fruits for the public eye .
**Hyp**   two men sit in a truck
Act **38.74**   True **contra**   Pred **contra**

**Pre**   outdoors in front of a crowd , a man plays an instrument by blowing into pipes he holds up to his face .
**Hyp**   a man sitting on the couch reading a book .
Act **34.87**   True **contra**   Pred **contra**

**Pre**   blurry people walking in the city at night .
**Hyp**   seven people dancing in a nightclub .
Act **34.41**   True **contra**   Pred **contra**

## Unit 280

`(((((NOT hyp:for) AND pre:NN) AND (NOT hyp:VB)) AND (NOT hyp:PRP)) OR overlap-75%)`

IoU **0.420**   w_entail 0.018      w_neutral -0.034      w_contra 0.022

**Pre**   the lady in the red jacket is helping the other lady decide what to buy .
**Hyp**   there are multiple people present .
Act **30.85**   True **entail**   Pred **entail**

**Pre**   a sports match is taking place between one team wearing the colors red and white and another team sporting the colors black and blue .
**Hyp**   the two teams are wearing different colors .
Act **28.93**   True **entail**   Pred **entail**

**Pre**   two men , one with a camera and another with hair clippers are helping another man in kitchen .
**Hyp**   three men are pictured
Act **28.62**   True **entail**   Pred **entail**

## Unit 283

`(((((NOT pre:and) AND hyp:IN) OR hyp:PRP$) AND hyp:.) OR hyp:VB)`

IoU **0.223**   $w_{entail}$ -0.086      $w_{neutral}$ 0.034      $w_{contra}$ 0.010

**Pre**      a soccer game with multiple males playing .
**Hyp**      a men 's soccer team winning the world cup .
Act **32.36**   True **neutral**   Pred **neutral**

**Pre**      a military group in uniform standing together while one of them gets their hat adjusted .
**Hyp**      a drill instructor is adjusted a students hat before they preform at a funeral .
Act **29.43**   True **neutral**   Pred **neutral**

**Pre**      a man is navigating a boat .
**Hyp**      a man is steering a large yacht down the lake .
Act **28.63**   True **neutral**   Pred **neutral**

## Unit 284

`(((((NOT hyp:JJ) AND overlap-25%) AND (NOT hyp:PRP$)) AND (NOT hyp:to) OR hyp:people)`

IoU **0.185**   $w_{entail}$ 0.033      $w_{neutral}$ 0.022      $w_{contra}$ -0.065

**Pre**      elegantly dressed in black , a man and woman embrace in dance .
**Hyp**      two people are dancing .
Act **26.72**   True **entail**   Pred **entail**

**Pre**      two men on bicycles competing in a race .
**Hyp**      people are riding bikes .
Act **22.49**   True **entail**   Pred **entail**

**Pre**      people walking to a special place .
**Hyp**      people are walking .
Act **21.08**   True **entail**   Pred **entail**

## Unit 302

`(((((hyp:for OR hyp:to) OR hyp:home) OR hyp:after) OR hyp:their)`

IoU **0.226**   $w_{entail}$ -0.053      $w_{neutral}$ 0.032      $w_{contra}$ 0.004

**Pre**      toddler walking along path .
**Hyp**      toddler is walking to his mom
Act **33.80**   True **neutral**   Pred **neutral**

**Pre**      uniformed schoolgirls are walking together on the street .
**Hyp**      the girls are walking home from school .
Act **30.83**   True **neutral**   Pred **neutral**

**Pre**      a pale dog runs down a path .
**Hyp**      a dog is running towards his owner
Act **28.92**   True **neutral**   Pred **neutral**

## Unit 362

`(((((hyp:outdoors OR hyp:outside) OR hyp:near) OR hyp:there) OR hyp:not)`

IoU **0.188**   $w_{entail}$ 0.041      $w_{neutral}$ -0.027      $w_{contra}$ -0.062

**Pre**      man and a woman walking on the street
**Hyp**      there are at least two people in the picture .
Act **36.00**   True **entail**   Pred **entail**

**Pre**      three women are sitting on a wharf and kicking their feet in the water .
**Hyp**      more than one person is touching a liquid .
Act **35.04**   True **entail**   Pred **entail**

**Pre**      a group of people playing guitars and singing .
**Hyp**      there are several people in this photo , and they are all making music .
Act **32.35**   True **entail**   Pred **entail**

## Unit 375

`(((((hyp:nobody OR overlap-75%) OR hyp:not) OR hyp:no) OR hyp:one)`

IoU **0.201**   $w_{entail}$ -0.007      $w_{neutral}$ -0.038      $w_{contra}$ 0.089

**Pre**      a band which includes an upright bass player is playing in a tent in front of canadian flags .
**Hyp**      the band has no bass player .

Act **30.21**   True **contra**   Pred **contra**

**Pre**        a boy with a concerned look it holding up two newspapers featuring a headline about murder .

**Hyp**        a boy is not holding anything .

Act **25.75**   True **contra**   Pred **contra**

**Pre**        a young boy wearing a red coat eats a chocolate bar .

**Hyp**        the boy has no clothes on .

Act **20.98**   True **contra**   Pred **contra**

---

## Unit 382

`(((((NOT hyp:there) AND hyp:NN) AND (NOT hyp:sitting)) AND (NOT hyp:standing)) OR hyp:VB)`

IoU **0.375**   w$_{entail}$ -0.022        w$_{neutral}$ 0.024        w$_{contra}$ 0.008

**Pre**        a group of people wearing hats and using walking sticks are walking through a wooded area on a trail .

**Hyp**        the tourists are being guided on their trip .

Act **40.71**   True **neutral**   Pred **neutral**

**Pre**        a gentleman in a striped shirt gesturing with a stick - like object in his hand while passersby stare at him .

**Hyp**        a gentleman in a striped shirt joyously gesturing

Act **40.40**   True **neutral**   Pred **neutral**

**Pre**        a middle - aged man in a gray t - shirt and brown pants sitting on his bed reading a flyer - like paper .

**Hyp**        he is reading a flyer about a new job he is interested in .

Act **40.04**   True **neutral**   Pred **neutral**

---

## Unit 386

`((((hyp:IN AND overlap-75%) OR hyp:not) OR hyp:no) OR hyp:only)`

IoU **0.198**   w$_{entail}$ -0.075        w$_{neutral}$ 0.014        w$_{contra}$ 0.060

**Pre**        a man in a blue shirt , khaki shorts , ball cap and white socks and loafers walking behind a group of people walking down a stone walkway with a water bottle in his left hand .

**Hyp**        a man in a blue shirt , khaki shorts , ball cap and blue socks and loafers walking behind a group of people walking down a stone walkway with a water bottle in his left hand .

Act **32.64**   True **contra**   Pred **contra**

**Pre**        a boy with a concerned look it holding up two newspapers featuring a headline about murder .

**Hyp**        a boy is not holding anything .

Act **26.37**   True **contra**   Pred **contra**

**Pre**        a dressed up woman walking next to a store at night .

**Hyp**        a dressed up woman is walking next to a pharmacy at night .

Act **25.10**   True **neutral**   Pred **neutral**

---

## Unit 390

`((((hyp:IN OR hyp:to) OR hyp:PRP$) AND (NOT hyp:EX)) OR hyp:NNP)`

IoU **0.422**   w$_{entail}$ -0.045        w$_{neutral}$ 0.033        w$_{contra}$ 0.010

**Pre**        two women walking in an area of UNK .

**Hyp**        two UNK workers walk down the street of the once beautiful suburban neighborhood , surveying the damage from the storm .

Act **41.84**   True **neutral**   Pred **neutral**

**Pre**        a group of kids are playing on a tire swing .

**Hyp**        a group of dogs are chasing a duck .

Act **39.84**   True **contra**   Pred **contra**

**Pre**        a woman walks by a brick building that 's covered with graffiti .

**Hyp**        the woman 's son drew some of the graffiti .

Act **37.97**   True **neutral**   Pred **neutral**