[Reviews · NeurIPS 2020]

Review 1

Summary and Contributions: The paper presents compositional explanations to automatically explain and characterize the behaviour of individual neurones in deep networks. Compared to existing approaches that rely on atomic concepts to explain neurones, the authors propose combinatorial compositions of logical forms to capture more explanations.

Strengths: The method is simple and general; it can be applied on many computer vision and NLP tasks. The authors provide thorough analysis about the relation between compositional explanations and model interpretability, performance and behaviour on image classification and natural language inference tasks. The experiments clearly show the significance of combinatorial compositions compared to the state-of-the-art which makes this work relevant.

Weaknesses: One of the main limitations I found in this approach, and it is clearly highlighted in the paper, is that to generate the proposed combinatorial concepts, semantic annotations are required. It is thus difficult to generalize to unseen examples if they have new concepts different from the training examples. The second issue is in the way logical forms were generated; the compositions are generated heuristically, this might limit finding better compositions of explanations, especially for NLP tasks. This limitation also makes me not confident about the reproducibility of the current work. Missing experiments with end-user for validating the broader claim.

Correctness: No obvious mistake.

Clarity: Very clear paper.

Relation to Prior Work: The proposed approach can be seen as an extension of NetDissect. The authors clearly mention the relation and differences between both works. The paper provides comparison results of both methods.

Reproducibility: No

Additional Feedback: - I appreciate that authors provided clear description about the limitations of their work which unveil some questions for future improvements. - It is clear from section 5. that explaining individual neurons doesn’t contribute too much to model accuracy, even in the case of positive correlations. The question do interpretable neurons contribute to model accuracy? may have two sides: No because individual neurones don’t contribute to model accuracy as shown by the explanations. Or No because the proposed approach can’t generate good explanations correlated with model’s decision. Could the authors clarify the confusion? -I am wondering how is simple to provide explanations and interpretations for complex network architectures and complex tasks such as object detection for example. I suggest doing some experiments on object detection tasks rather than simple classification and see if compositional explanations can help interpreting the relation between class scores and bounding box locations. ** Acknowledge of authors feedback


Review 2

Summary and Contributions: ==================AFTER AUTHOR RESPONSE====================== The authors seem to have satisfactorily answered my questions and state that they'll include these analyses in the supplement. ======================INITIAL REVIEW============================= Summary This paper proposes a way to explain neural network models as a composition of atomic concepts based on observed neuron activations. This paper considerably extends previous work (on NetworkDissection) by allowing concepts to be composed by defining logical operators, and applying a beam search to identify a logical form that closely matches the neuron behaviour. They apply their procedure to study models trained on an image classification (scene classification) task and a Natural Language Inference task. Contributions: 1. A procedure to explain neurons (in a trained model) as a composition of logical concepts. 2. They answer interesting questions regarding the interpretability of vision and language models. a) the kind of abstractions learned by neurons (e.g. that neurons can be triggered by abstract concepts that can be a lot more specific e.g. “water, but not blue”) b) compositional explanations as a lens for analyzing model performances in vision and language models, and in particular how models in these 2 domains behave differently. 3. They also show how compositional explanations can be used to generate adversarial examples to change the model output in a predictable manner.

Strengths: 1. The idea of mapping neurons to compositional logical concepts is interesting, and the paper provides a nice mechanism to do it. It is also written clearly. 2. There are a lot of details in terms of getting the implementation right for explaining the NLI models, and those choices are explained clearly. Hopefully code will also be made available. 3. The paper provides a good way to measure the effectiveness of the compositional explanations i.e. quantitative evaluation (in Fig. 3). 4. They help answer interesting questions about how explanation/interpretability can be used to analyze model behaviour, and demonstrate it on models in language and perception domains.

Weaknesses: 1. It would have been interesting to see the variety in the types of compositional logical concepts identified by the models used in this study. e.g. a. Are some neurons identifying similar compositional concepts - to what extent are they similar or dissimilar (e.g. maybe at length 3, k neurons are similar, but by length 6 all of them are different). Is there some way of saying how granular these concepts are? b. Which are the most frequent compositional concepts that are shared by some neurons? c. Are different visual models identifying the same or similar compositional concepts?

Correctness: Yes. As mentioned there are a lot of details in terms of getting the implementation right for explaining the NLI models, and those choices are explained clearly. Hopefully code will also be made available.

Clarity: Yes

Relation to Prior Work: Yes. Substantially improves previous work in interesting way by adding compositionality to concepts that can be used to map to neuron/model behavior.

Reproducibility: Yes

Additional Feedback: Figure captions can be improved a bit more, e.g. Fig 8, it would be good to associate the 3 figures in the bottom with the 3 neurons that get fired.


Review 3

Summary and Contributions: The study proposes a generalization of an interpretation method of deep representations (Network Dissection) by performing a search not only over primitive concepts but also over bounded logical compositions of them. The interpretation method is explored with two tasks, from vision and from language processing. The authors then assess correlations between model performance and the interpretations of single-neuron activations. Finally, the authors test causality by manipulating inputs based on the conceptual interpretations and testing the effect on model prediction. Contributions - A generalize framework for identifying conceptual explanations of single-unit activtations in deep representations. - A method to test model biases by generating adverserial examples based on learned explanations. I have read the rebuttal and comments from the other reviewers and I retain my judgment that this paper should be accepted for publication.

Strengths: The theoretical framing is simple and clear, and the claims are sound and supported by empirical tests.

Weaknesses: Novely is limited since it is a generalization of a well-known interpretation method, however, the implementations and results are nonetheless interesting and I therefore find the contriubtion of the paper significant.

Correctness: The application of the methods and analyses seem correct and well performed.

Clarity: Very well written.

Relation to Prior Work: Clear discussion and relevant refs to previous works.

Reproducibility: Yes

Additional Feedback: A suggestion - figure+Legend are not always self-explanatory. It would be practical to have all info required for understanding the figure in the legend (possibly moved from the main text). For example, in fig8 it would be good to explain that the left image was presented to the model, the right one is the 'copy-pasted' image, and that below are sample images from the new prediction. Regarding the copy-paste example, how sensitive would be the results to varying positions/sizes of the inserted sub-image?


Review 4

Summary and Contributions: This paper presents a quite thoughtful and informative attempt to understand what information is encoded in individual neurons and sets of neurons (by which the authors mean units in neural networks, not real brain neurons. This is a somewhat unfortunate use of the word and it might be helpful in the future to talk on neural network units or some other term instead.) It is by now clear to anyone working with these networks that each individual neuron, esp. at the lower and intermediate levels, encodes not some simple human-understandable feature (like color=red, size=large, wordclass=noun, type=Person), but rather a complex combination of what one could call sub-facets, which each by itself often not easily described to a human. Combinations of these sub-facets taken from different neurons acting in tandem JOINTLY encode the facets that are more accessible to humans. But of course the other sub-facets also encoded by the neurons present in a human-accessible feature cluster might encode [parts of] a variety of totally unrelated other features, with the result that simple hotspot analysis and similar highlighting techniques are never fully determinate or clear, but always rather ambiguous and ‘smeared’. One can treat a combination of sub-facets that form a feature as a ‘decision rule’ as stated on line 32, but the idea of rather focusing on the compositionality of sub-facets to form features and even higher-level feature clusters (= concepts, like dog and human and happiness) is a good approach. The question is: can one make it actually work? Can one define and actually build composition operator that work over ‘primitive concepts’, or features, or sub-facets?

Strengths: The method proposed in the paper is fairly obvious, given all this background. In an image classification task, identify the sets of neurons that reliably correlate or anti-correlate with a human human-identifiable feature, and define their combination to be such an operator. Test this by first presenting targets for processing that activate such identifiable sets, and then overlaying onto the target the appropriate opposite features to demonstrate that the identified neuron combination (but no other neurons) is affected. Then you have it: an identifiable feature is uniquely correlated with an identified neuron set. You can then find combinations of them (using AND, OR, NOT, etc.) for other features. Finally you can use this mapping to provide ‘explanations’ for new input: whenever the identified neuron set is active, you state that the feature has been seen and affects the outcome. This idea is described in Section 2, with the addition of a step that discretizes the continuous neuron values using binary masks. The two tasks are described in Section 3. I think they are reasonably well chosen but NLI as a representative for text processing is problematic. Because inference is a complex question, the NLI corpora are not very well design and built, and therefore provide much less internally consistent and trustworthy indicators of semantics than some other corpora (for example, coreference, or semantic type assignment). Sections 4 (on image classification) and 5 (on NLI) are interesting but unfortunately too short to provide adequate details needed for this complex question.

Weaknesses: Section 6 could itself be an entire paper. It provides tantalizing examples, but no large-scale measurements, tables, or graphs. So it’s not clear what one should make of this section.

Correctness: It is hard to judge. The paper is certainly interesting, more for the approach than the actual implementations.

Clarity: Reasonably

Relation to Prior Work: Yes

Reproducibility: Yes

Additional Feedback: Overall, the paper addresses an extremely important and difficult question in an imaginative, thoughtful, and somewhat original way. I like this very much. But it tackles much more than it can deliver and that is a pity. Could you add more concrete results somehow? Still, think the paper should be considered seriously for acceptance as a thought-initiator. It is far richer than just another network-optimization hack for better scores.

[Author Response · NeurIPS 2020]

Thanks to all for thoughtful and helpful comments, and positive feedback! Reviewers agree we have proposed a "simple and general" (**R1**) yet "imaginative and thoughtful" (**R4**) method which tackles an "extremely important" problem (**R4**) and produces significant, interesting (**R2**, **R3**), and "thought-initiat[ing]" (**R4**) results. We now address specific points:

**R1: Our method requires (hand-labeled) semantic annotations.** Not necessarily: in the natural language inference (NLI) experiments, we use a pretrained model to label the probe dataset with part-of-speech tags. One could also use a semantic segmentation network to generate visual concepts. Of course, these models must be trained on annotated data—at some point, any procedure for labeling neurons with concepts requires some starting source of labeled concepts.

**R1: Heuristic generation of concepts is a limitation and hinders reproducibility.** We agree that the choice of primitive concepts and compositions highly influence the discovered concepts, and that beam search only approximates the preferable, but intractable, enumeration over logical forms. However, we disagree that this hinders the *reproducibility* of our results; as **R2** notes, we have tried to precisely specify our set of inputs, concepts, and models, especially for NLI. We will also release code with the camera-ready paper, which should ease reproducibility concerns (cc **R2**).

**R1: We may not find a strong relationship between interpretability and accuracy if we don't generate the right explanations.** This is true, because "interpretability" is defined by the space of concepts we specify. If we find no relationship between interpretability and accuracy in a model for an explored set of concepts, it could be that we are simply using the wrong definition of "interpretability", and that alternative concept spaces could lead to more informative results. We show two tasks where we discover interpretable concepts with a noisy, but still highly significant, correlation with accuracy.

**R4: NLI isn't the best task to explore semantics, since NLI datasets are poorly built.** Indeed, this is precisely why we chose NLI: since previous work has shown that NLI models learn non-robust, shallow heuristics, our experiments explore how these strategies are implemented in individual neurons.

**R2: Do neurons identify similar compositional concepts? What about different models?** Great questions! Figure S1 plots the counts of each concept across the 512 units of ResNet-18, by length. At length 1 (NetDissect), many concepts appear multiple times; the mean number of occurrences per concept is 2.61 (42% of concepts are unique). Uniqueness increases dramatically by length 3 (mean 1.03; 97% unique), 5 (1.01; 99%), and 10 (1.00; 100%). Our explanations thus reveal significant specialization in neuron function (vs. NetDissect). Table S1 shows some repeated concepts. We will add this to the supplement and analyze NLI as well (omitted here for space). We leave the question of different vision models for future work; our code will facilitate the necessary experiments. The adversarial examples in Figure 8 hint that some concepts (e.g. *non-blue water*) are shared across models.

**R3: How sensitive are copy-paste examples to size and position?** In Figure S2 we vary the size and position of subimages for the copy-paste examples (note this analysis is less straightforward for examples like *non-blue water*). Sensitivity depends on the specific example. In general, if the sub-image is too small (left), the original class prevails; otherwise, the *igloo → clean room* example is quite reliable, while the *street → fire escape* example is less so. We will add this to the supplement.

**R1: Why not object detection?** As noted by the original NetDissect work, for networks explicitly trained on object detection tasks, it would be less surprising that neurons specialize for object detection. This motivates us to explore a scene recognition network, and see whether or not interpretable object-level concepts emerge without explicit object-level supervision. Still, probing object detection models is an interesting and straightforward extension of our method.

Figure S1: Number of unique concepts based on length.

Figure S2: Varying the size and position of sub-images. Green: prediction changes to intended adversarial class; yellow: prediction changes to a different class (e.g. *aqueduct* for the middle row); red = no change.

Table S1: Most common concepts by length $N$

| $N$ | Concept | # |
| --- | --- | --- |
| 1 | pool table | 15 |
| | house | 12 |
| | corridor | 11 |
| 3 | pillow OR (bed AND bedroom) | 4 |
| | sink OR toilet OR bathtub | 3 |
| | water OR river AND (NOT blue) | 2 |
| 5 | auditorium OR theater OR conference center... | 2 |

[Meta-Review · NeurIPS 2020]

The reviewers were quite positive about this paper, which works to automatically explain the behavior of neurons in deep networks using compositionality. These results will be of interest to both those in NLP and computer vision. The reviewers raised a few points for clarification (requiring labeled data, choice of tasks), which were handled in the rebuttal and the authors state they will incorporate these caveats and clarifications into the final version and/or supplementary material.